# Enforcing 3D Topological Constraints in Composite Objects via Implicit Functions

## Abstract

Medical applications often require accurate 3D representations of complex organs with multiple parts, such as the heart and spine. Their individual parts must adhere to specific topological constraints to ensure proper functionality. Yet, there are very few mechanisms in the deep learning literature to achieve this goal.

This paper introduces a novel approach to enforce topological constraints in 3D object reconstruction using deep implicit signed distance functions. Our method focuses on heart and spine reconstruction but is generalizable to other applications. We propose a sampling-based technique that effectively checks and enforces topological constraints between 3D shapes by evaluating signed distances at randomly sampled points throughout the volume. We demonstrate it by refining 3D segmentations obtained from the nn-UNet architecture.

## 1 Introduction

Many medical applications require representing the 3D shapes of complex organs made of several parts, such as the four chambers of the heart and the vertebrae that compose the spine. These individual parts must meet topological constraints to ensure proper functionality. For example, in the human heart, the left ventricle and the myocardium must touch each other—but not overlap—with specific contact surface ratios[1] Buckberg et al. (2018), as shown in the top row of Fig. 1. Conversely, in the human spine, adjacent vertebrae must maintain a gap between them—where joint capsules are placed—as shown in the bottom row of the figure.

Such constraints on contact ratios or gaps between organ components are known *a priori* from centuries of medical practice and do not need to be learned from data. Thus, these constraints should be explicitly enforced to create models with correct topology, which is crucial for accurate modelling and downstream analyses. Yet, to the best of our knowledge, there are no formal mechanisms in the deep learning literature to do so. Existing constraint-satisfaction mechanisms mainly focus on preventing intersections between object parts Vasu et al. (2022); Ye et al. (2022); Ma et al. (2020); Hassan et al. (2020) or ensuring containment across different categories Gupta et al. (2022). Constraint violations are typically detected and resolved locally, often using a sliding-window approach Gupta et al. (2022). However, enforcing more complex constraints, such as those described above, cannot be done in this manner. For instance, for heart reconstruction, calculating the ratio between the area of the contact and object surfaces cannot be done locally. Similarly, for spine reconstruction, checking the gap between two objects becomes infeasible if the gap size exceeds the sliding window step.

In this paper, we propose using deep implicit signed distance functions Park et al. (2019a) (SDFs) to enforce topological constraints between 3D object parts. Even though we focus on 3D human heart and spine reconstruction, the techniques we introduce are generic and could be used in many other scenarios. We demonstrate that using implicit function allows for effective checking and enforcement of topological constraints between 3D shapes. The core idea is to observe and resolve potential constraint violations via a Monte Carlo approximation method: We sample many random points throughout the entire volume and calculate their signed distances to all object parts. These distances reveal how objects are connected to each other and if certain constraints are violated. From this, we can identify points that indicate constraint violations and adjust the signed distance

---

[1]Contact Ratio = (Area of the contact surface) / (Sum of the areas of both surfaces)

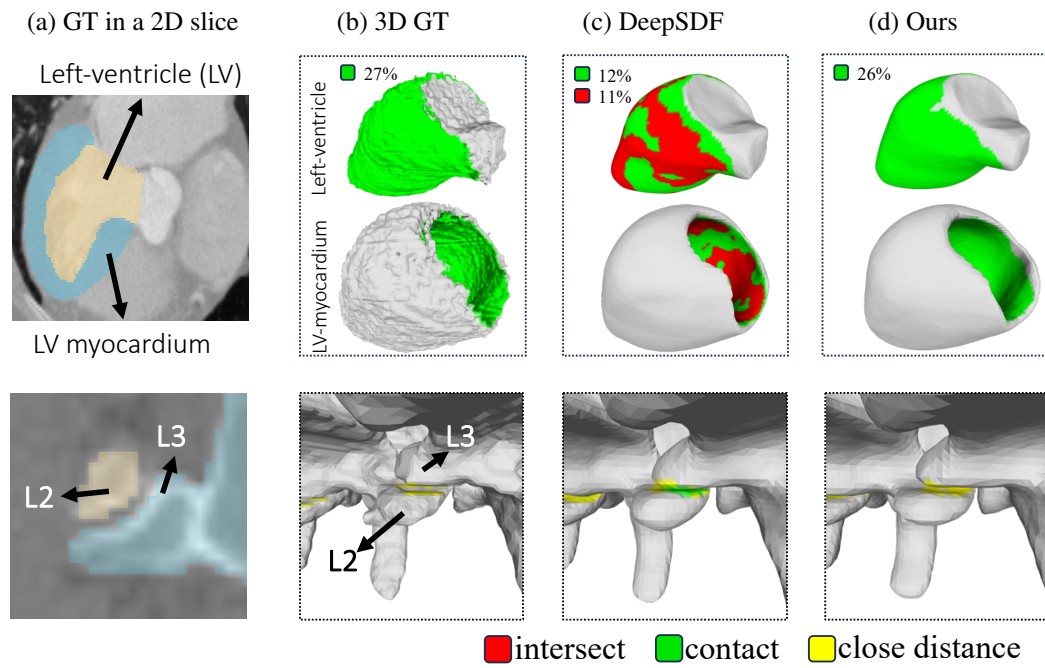

Figure 1: **Topological Constraints in Composite Organs.** (a) Ground-truth in a 2D MRI slice for the left ventricle and its myocardium at the top, and vertebrae L2 and L3 at the bottom. (b) Ground-truth in 3D. In the top row, we show the same volume twice, once with only the left-ventricle and second time with only the myocardium for better visibility of the contact areas. The contact ratio between the two is 27% and that remains fairly constant across people. In the bottom row, there must be a minimum gap between L2 and L3. (c) The output surfaces when fitting DeepSDF Park et al. (2019a) to the nn-UNet Isensee et al. (2018) output and (d) when using our approach to fit the same nn-UNet output. In (c,d), green denotes contact, red - intersection, and yellow - proximity. For the myocardium, using DeepSDF yields intersections and the contact areas are random, whereas in our case there are no intersections and the contact area is properly shaped. For the vertebrae, we eliminate the contacts and areas of close proximity are properly modeled.

functions at those points to resolve the constraints accordingly. Repeating this process with enough random points yields constraint-compliant composite shapes. In practice, we typically use 300K points, resulting in a 30% computational overhead.

In the heart reconstruction case, given two objects represented by their SDFs and a prior indicating a desired contact ratio (%) between them, our goal is to refine their 3D shapes so that they do not intersect and contact each other with that exact ratio. We first uniformly sample a large number of points and compute the signed distances between them and the two surfaces. At any given state of the objects, we can estimate the contact ratio by counting the number of random points that are in close proximity to both objects. Similarly, in spine reconstruction, we modify the surfaces at points where the sum of distances to both surfaces is smaller than a threshold to ensure there is a proper gap between them.

Our implicit functions are trained solely on 3D training shapes. At test time, we use the resulting latent vector models to refine the 3D segmentations obtained from the popular nn-UNet segmentation architecture Isensee et al. (2018). Our experimental results show that simply fitting the implicit functions to nn-UNet segmentations results in 3D composite shapes that are not topologically meaningful, as object parts may penetrate each other and fail to follow prior constraints. In contrast, our sampling-based method produces 3D composite shapes with proper connectivity between object parts, adhering to the specified constraints, and achieving lower reconstruction error than the original nn-UNet segmentations.

In summary, our contribution is an easy-to-implement and effective approach to using implicit functions to enforce topological consistency while modeling complex multi-part objects. This is key to obtaining accurate, medically useful results.

## 2 RELATED WORK

**Image-Based Topological Interaction.** Many 2D segmentation problems involve semantic classes with relative topology constraints between them, such as road connectivity over a background or cell nuclei that should be contained within the cytoplasm. For example, a topology-aware loss that uses the response of selected filters from a pre-trained VGG19 network is introduced in Mosinska et al. (2017). These filters prefer elongated shapes and promote better connectivity. Similarly, a topology loss based on persistent diagrams is used in Hu et al. (2019) to improve cell segmentation. Other methods rely on detecting and penalizing critical pixels for topology interaction between classes, such as Hu (2021) and Gupta et al. (2022), using homotopy warping and convolutions respectively to find these pixels. However, image and pixel based topology constraints do not lend themselves naturally to implicit multi-object 3D reconstruction.

**Multi-Object 3D Reconstruction.** It is a fundamental task for scene understanding or generation Irshad et al. (2022a); Liu & Liu (2021). The presence of multiple objects poses a different set of challenges compared to single-object reconstruction where objects are usually treated as isolated geometries without considering the scene context, such as object locations and instance-to-instance interactions. For multi-object reconstruction, Mesh R-CNN Gkioxari et al. (2019) augments Mask R-CNN He et al. (2017) with a mesh predictions branch that estimates a 3D mesh for each detected object in an image. Total3DUnderstanding Nie et al. (2020) presents a framework that predicts room layout, 3D object bounding boxes, and meshes for all objects in an image based on the known 2D bounding boxes. However, these three methods first detect objects in the 2D image, and then independently produce their 3D shapes with single object reconstruction modules. Liu & Liu (2021) proposes a system to infer the pose, size, and location of 3D bounding boxes and the 3D shapes of multiple object instances in the scene, which is divided into a grid whose cells are occupied by objects. Irshad et al. (2022b) recovers objects shape, appearance, and poses using implicit representations, as has become increasingly frequent, *e.g.*, Irshad et al. (2022b); Karunratanakul et al. (2020); Ye et al. (2022); De Luigi et al. (2023). For the most part, these works focus on reconstruction accuracy while our work focuses on the correctness of the topological constraints between components.

**3D Interactions.** Some methods try to enforce consistency between objects in a scene when performing a 3D reconstruction, as in Engelmann et al. (2020) where a collision loss between reconstructed objects is minimized. However, most such efforts focus on human-object interactions. In Karunratanakul et al. (2020), the grasp between the hand and an object is modeled in terms of implicit surfaces and the algorithm learns to generate new grasps using a VAE. The method of Ye et al. (2022) jointly reconstructs the hand-object interaction from an image and the object as an implicit surface. Contact2Grasp Li et al. (2023) learns to synthesize grasps by first predicting a contact map on the object surfaces. Other research directions include body-garment interaction, such as DrapeNet De Luigi et al. (2023) with a physically based self-supervision, or human-scene interaction Hassan et al. (2020). All these works primarily focus on interactions between the articulated human body and objects, without incorporating any prior information on how they should interact. In contrast, our approach considers a specific topological and geometric constraint that needs to be hold between objects.

## 3 METHOD

Our method refines composite implicit 3D shapes to ensure that their individual parts strictly meet predefined constraints. We focus on two different kinds of constraints—neither of which has been considered in previous work—in two distinct scenarios. First, when reconstructing the four chambers of the human heart, these chambers should never intersect but instead should be in contact with each other over a given percentage of their surface areas. For example, in most people, the left ventricle surface and its myocardium exhibit a 27% contact ratio, that is, the area of the contact surface divided by the sum of the areas of both surfaces. Such contact ratios are consistent across all annotated instances and reflect the topological relationship between internal parts of the heart. Thus, a proper 3D reconstruction should result in shapes that exhibit these contact ratios. Second, when reconstructing a healthy human spine, there should be a gap between adjacent vertebrae of at least 1 mm Little & Khalsa (2005). This constraint can be violated due to medical conditions such as

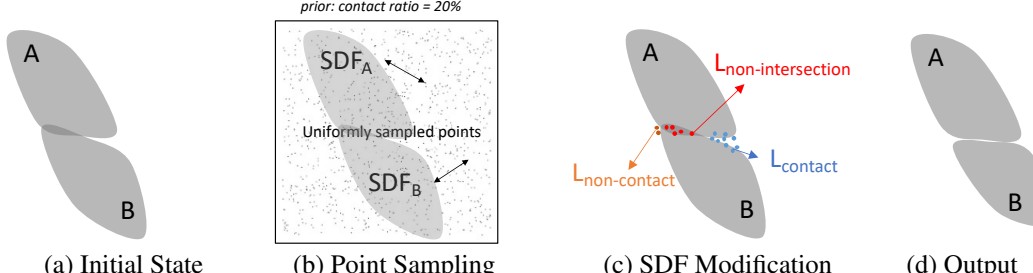

(a) Initial State     (b) Point Sampling     (c) SDF Modification     (d) Output

Figure 2: **Method overview.** (a) We start with shapes defined by their respective SDFs Park et al. (2019b) and parameterized by a latent vector. (b) We uniformly sample points around them. (c) We select a subset of topologically meaningful points such as ones residing close the object surfaces and use them to write loss functions. (d) We minimize these loss functions with respect to the latent vectors to enforce the constraints.

lumbar facet arthritis Ergan et al. (1997); Mann & Singh (2019). However in this paper, we restrict ourselves to healthy subjects for whom this constraint must be satisfied.

In these experiments, we apply our approach to achieve either specified contact ratios or to enforce minimum distance constraints between objects. These two constraints are fundamentally different and present distinct challenges. For contact ratios, the primary difficulty lies in ensuring that objects interact without penetrating each other, often requiring precise alignment and continuous adjustment of surface boundaries. For minimum distance constraints, detecting violations necessitates global checks to ensure that no parts of the objects come too close from each other. Addressing these two distinct sets of requirements using traditional modeling frameworks is often cumbersome and requires separate, specialized approaches for each case. In contrast, our approach handles both scenarios in a consistent, unified manner.

As we use DeepSDF Park et al. (2019a) to model individual parts and corresponding latent vectors to parameterize them, we first provide a brief overview of the DeepSDF method. Next, we describe our method for enforcing the contact ratio constraint in Sec.3.2 and then show how the same methodology is applicable for enforcing the minimum distance constraint in Sec.3.3. In both cases, we refine all the object parts simultaneously by updating their latent vectors. We uniformly sample points around them and identify a subset of topologically meaningful points, particularly those near the object surfaces, to formulate our loss functions. We then minimize these loss functions with respect to the latent vectors to enforce the constraints, as summarized in Fig.2.

### 3.1 DEEP SDF

SDFs have emerged as a powerful model to learn continuous representations of 3D shapes. They allow detailed reconstructions of object instances as well as meaningful interpolations between them. Given an object, a signed distance function outputs the point's distance to the closest object surface. We write as:

$$f(\mathbf{z}, \mathbf{x}) = s : \mathbf{x} \in \mathbb{R}^3, s \in \mathbb{R} \,, \tag{1}$$

where $\mathbf{z}$ is a latent vector that parameterizes the surface. Conventionally, the distance is negative if the point is inside the object and positive otherwise. By varying $\mathbf{z}$, it becomes possible to deform the object so that it can represent any shape within a category, such as a left or right ventricle in our case. As in Park et al. (2019a), we implement $f$ using a multi-layer perceptron whose weights are learned using an auto-decoding method on a large set of instances of a specific type.

### 3.2 ENFORCING CONTACT RATIOS

We now describe our approach to enforcing contact ratio constraints: Given two objects represented by their SDFs and a prior indicating a desired contact ratio (%) between them, our goal is to refine their 3D shape in such a way that they do not intersect and contact each other with that exact ratio. In our experiment, we reconstruct simultaneously five components of the human heart: the four ventricles and the myocardium of the left-ventricle where the contact ratio between all pairs among these five components are given.

### 3.2.1 MINING TOPOLOGICALLY MEANINGFUL POINTS

We refine the deep implicit functions described above to obey the contact ratio constraint. To do so, we sample the space uniformly and identify three sets of topologically meaningful points:

- $\mathcal{A}_{\text{contact}}$ : points that should be in contact with both objects;
- $\mathcal{A}_{\text{intersecting}}$ : points that sit inside both objects;
- $\mathcal{A}_{\text{non-contact}}$ : points that should not be in contact with both objects.

Given these sets, we design loss functions whose minimization ensures the constraint are met, as summarized in Fig. 2.

Let us first consider the simple case of two *explicit* surfaces $A$, $B$. The contact ratio between $A$ and $B$ is defined as:

$$P_{A,B} = \frac{\text{Area}(S_{AB})}{\text{Area}(S_A) + \text{Area}(S_B)} \ , \tag{2}$$

where $S_{AB}$ represents the partial surface of object $A$ that is in close proximity, i.e., contact distance, to object $B$, and $S_A$ and $S_B$ refer to the entire surfaces of objects $A$ and $B$, respectively.

When $A$ and $B$ are represented as implicit signed distance functions $f_A$ and $f_B$ and latent vectors $\mathbf{a}$ and $\mathbf{b}$ instead of explicit meshes, we show that the contact ratio can be approximated by

$$P'_{A,B} = \frac{\sum_{i=1}^{N} \mathbb{1}(|f_A(\mathbf{a}, \mathbf{x}_i)| < \epsilon) \cdot \mathbb{1}(|f_B(\mathbf{b}, \mathbf{x}_i)| < \epsilon)}{\sum_{i=1}^{N} \mathbb{1}(|f_A(\mathbf{a}, \mathbf{x}_i)| < \epsilon) + \sum_{i=1}^{N} \mathbb{1}(|f_B(\mathbf{b}, \mathbf{x}_i)| < \epsilon)}, \tag{3}$$

where $\{\mathbf{x}_i : i \in (1, N)\}$ is a set of $N$ uniformly sampled random points, $\epsilon$ is a small threshold indicating the contact distance, and $\mathbb{1}(\cdot)$ is the indicator function that returns the value 1 if its statement is true and 0 otherwise. In essence, the implicit contact ratio is estimated via a stochastic Monte Carlo by counting the number of points lying close to the surfaces of both objects and the number of points lying close to the surface of either one.

Our goal is to modify $\mathbf{a}$ and $\mathbf{b}$ so that the estimated contact ratio matches the correct one ($\text{Prior}_{A,B}$). To do so, we focus on altering the numerator of Equation 3, which represents the implicit contact surface. Note that theoretically it is also possible to modify $\mathbf{a}$ and $\mathbf{b}$ to influence the denominator of Equation 3. However, in practice, this denominator, reflecting the combined surface areas of the two objects, tends to remain relatively constant during optimization. This is because the overall sizes and shapes of the objects are mainly determined by the initial segmentation inputs. Thus, we take the expected number of points that should be proximal to both A and B to be

$$N_{\text{contact}} = \text{Prior}_{A,B} \times \left( \sum_{i=1}^{N} \mathbb{1}(|f_A(\mathbf{a}, \mathbf{x}_i)| < \epsilon) + \sum_{i=1}^{N} \mathbb{1}(|f_B(\mathbf{b}, \mathbf{x}_i)| < \epsilon) \right) \tag{4}$$

where $\text{Prior}_{A,B}$ is the prior contact ratio between the two categories that A and B belong to. Given $N_{\text{contact}}$, we define two sets of points we will use to write loss functions.

- $\mathcal{A}_{\text{contact}}$ : The top $N_{\text{contact}}$ points with closest summed distance to both objects. The summed distance from a point $\mathbf{x}$ to the two objects is defined as: $|f_A(\mathbf{a}, \mathbf{x}) + f_B(\mathbf{b}, \mathbf{x})|$.
- $\mathcal{A}_{\text{non-contact}}$ : All points that are in close proximity to both $f_A$ and $f_B$ but are not included in $\mathcal{A}_{\text{contact}}$ : $\mathcal{A}_{\text{non-contact}} = \{\mathbf{x} \mid f_A(\mathbf{a}, \mathbf{x}) < \epsilon \wedge f_B(\mathbf{b}, \mathbf{x}) < \epsilon \wedge \mathbf{x} \notin \mathcal{A}_{\text{contact}}\}$

In addition, we find the points that are within both objects

- $\mathcal{A}_{\text{intersecting}} = \{\mathbf{x} \mid f_A(\mathbf{a}, \mathbf{x}) < 0 \wedge f_B(\mathbf{b}, \mathbf{x}) < 0\}$.

### 3.2.2 COMPATIBILITY LOSSES

Let us consider two objects of categories $A$ and $B$, each with its associated SDF $f_A$ and $f_B$, defined as in Eq. 1. In what follows, the perceptrons implementing $f_A$ and $f_B$ are trained and their weights are frozen. Thus, all refinements of the objects are obtained by optimizing loss functions with respect to the latent vectors $\mathbf{a}$ and $\mathbf{b}$ that parameterize the objects. We periodically find those topologically meaningful points, as described in the previous section, and use them to modify implicit surfaces using compatibility losses defined in this section.

**Self Intersection Loss.** To prevent intersections between $A$ and $B$, we find all anchor points that lie within both objects — specifically, points where both $f_A$ and $f_B$ are negative. We then formulate the loss based on these points:

$$\mathcal{L}_{\text{intersecting}} = \sum_{\mathbf{x} \in \mathcal{A}_{\text{intersecting}}} |f_A(\mathbf{a}, \mathbf{x})| + |f_B(\mathbf{b}, \mathbf{x})| \, . \tag{5}$$

Minimizing this loss pushes the surfaces of $A$ and $B$ towards those points and discourages self-intersections.

**Contact Ratio Loss.** To enforce a given contact ratio between surfaces, we must first enforce all points in the $\mathcal{A}_{\text{contact}}$ to be proximal to both surfaces by minimizing the loss:

$$\mathcal{L}_{contact} = \sum_{\mathbf{x} \in \mathcal{A}_{\text{contact}}} |f_A(\mathbf{a}, \mathbf{x}) + f_B(\mathbf{b}, \mathbf{x})|. \tag{6}$$

Minimizing this loss pushes the two implicit surfaces toward those points without overlap or penetration. The loss function reaches its minimum when $f_A(\mathbf{a}, \mathbf{x}) = -f_B(\mathbf{b}, \mathbf{x})$. In this case, objects $A$ and $B$ are equidistant to the anchor point $\mathbf{x}$, which must reside within at most one of them, or precisely on the surface of both in the scenario where $f_A(\mathbf{a}, \mathbf{x}) = f_B(\mathbf{b}, \mathbf{x}) = 0$.

Then, to ensure there is no more than $N_{\text{contact}}$ points in proximal to both A and B, we minimize the loss function:

$$\mathcal{L}_{non-contact} = \sum_{\mathbf{x} \in \mathcal{A}_{\text{non-contact}}} -(f_A(\mathbf{a}, \mathbf{x}) + f_B(\mathbf{b}, \mathbf{x})). \tag{7}$$

**Data Loss.** To ensure the refined surfaces match the heart outlines, we sample a set $\mathbf{X}$ of SDF samples for each object based on its segmentation, which consists of 3D points $\mathbf{x}$ and their SDF values $s_{\mathbf{x}}$. Similar to DeepSDF Park et al. (2019a), we sample 500000 spatial points per object and sample more aggressively (90%) near the object surface. The data loss is defined as:

$$\mathcal{L}_{data} = \sum_{\mathbf{x} \in \mathbf{X}_A} |f_A(\mathbf{a}, \mathbf{x}) - s_{\mathbf{x}}| + \sum_{\mathbf{x} \in \mathbf{X}_B} |f_B(\mathbf{b}, \mathbf{x}) - s_{\mathbf{x}}|. \tag{8}$$

### 3.2.3 OPTIMIZATION

Given these losses, we reconstruct the two 3D objects simultaneously by minimizing the joint loss function:

$$\mathcal{L} = \mathcal{L}_{\text{intersecting}} \times \lambda_1 + \mathcal{L}_{\text{contact}} \times \lambda_2 + \mathcal{L}_{\text{non-contact}} \times \lambda_3 + \mathcal{L}_{\text{data}} \times \lambda_4, \tag{9}$$

where $(\lambda_1, \lambda_2, \lambda_3, \lambda_4)$ are controlling parameters. During the optimization process, we periodically sample 300K points after each 10 iterations to update the set of topological meaningful points. This is because the objects are continuously refined throughout this optimization, and changes in object shapes would result in different sets of topologically meaningful points.

### 3.3 ENFORCING MINIMUM DISTANCE CONSTRAINTS

Similarly to the contact ratio case, we uniformly sample points in the entire volume and identify a set of topologically meaningful points and use them to formulate loss functions to adjust the surfaces accordingly. Given two objects represented by their implicit signed distance functions $f_A$ and $f_B$ and parameterized by the latent vectors $\mathbf{a}$ and $\mathbf{b}$, respectively, our goal is to refine the two objects so that the minimum distance between them exceeds a given prior threshold $d$. In this case, we are interested in a set of topologically meaningful points $\mathcal{A}_{\text{violation}}$ where the sum of distances to both surfaces is smaller than $d$. We take it to be

$$\mathcal{A}_{\text{violation}} = \{\mathbf{x} | (f_A(\mathbf{a}, \mathbf{x}) + f_B(\mathbf{b}, \mathbf{x})) < d\} \, . \tag{10}$$

Then, we modify the signed distance functions at those points by minimizing

$$\mathcal{L}_{\text{min-distance}} = \sum_{\mathbf{x} \in \mathcal{A}_{\text{violation}}} \max(0, d - (f_A(\mathbf{a}, \mathbf{x}) + f_B(\mathbf{b}, \mathbf{x})) \tag{11}$$

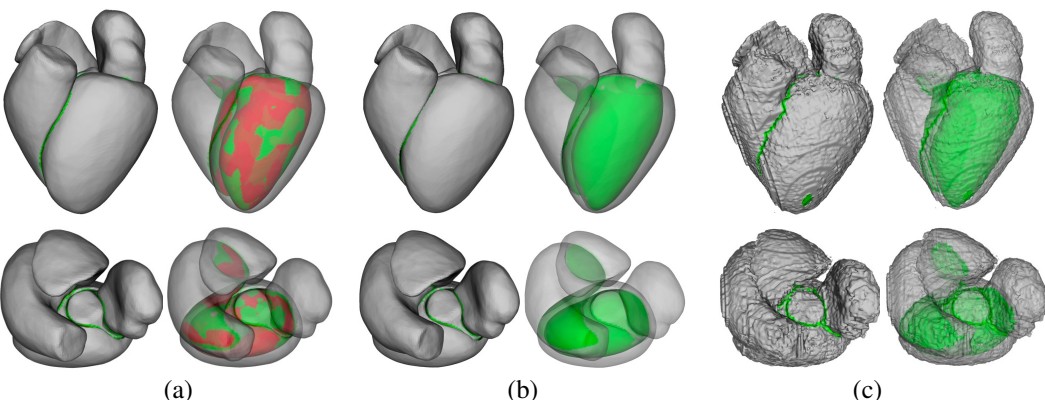

(a)             (b)             (c)

Figure 3: **Heart Reconstruction from an In-Distribution sample.** The two rows depict the same heart as seen from two different viewpoints. In each case, we show the opaque shape on the left and a transparent version to reveal the contact areas on the right. Red indicates penetration while green denotes proper contact. (a) Fitting SDFs individually to each component. There are many red areas. (b) Fitting SDFs jointly and imposing constraints using our method. The red areas have disappeared and the contacts are now correct. (c) Ground-truth.

to push both surfaces away from the points in $\mathcal{A}_{\text{violation}}$.

Then, similarly to the contact ratio scenario in Sec.3.2, we reconstruct all 3D objects together and enforce the constraint simultaneously by minimizing the joint loss function

$$\mathcal{L} = \mathcal{L}_{\text{min-distance}} \times \lambda_1 + \mathcal{L}_{\text{data}} \times \lambda_2 \, , \tag{12}$$

where $(\lambda_1, \lambda_2)$ are controlling parameters.

## 4 EXPERIMENTS

We demonstrate the effectiveness of our method on two key use cases: 3D whole-heart reconstruction and lumbar spine reconstruction, where we fit deep implicit functions Park et al. (2019a) to segmentation outputs from nn-Unet Isensee et al. (2018). Our method jointly reconstructs all object parts while enforcing their topological constraints between them. For comparison purposes, we use a baseline method that fits each part individually. We will show that our method produces meaningful, topologically accurate shapes in all cases, a result that cannot be achieved using the baseline. Compared to our method, nn-Unet struggles to generalize to out-of-training-distribution heart images and fails to produce topologically accurate lumbar spines.

**3D Whole-Heart Segmentation.** We use our method to simultaneously reconstruct five human-heart components: Left ventricle (LV), myocardium of left ventricle (M-LV), left atrium (LA), right atrium (RA), and right ventricle (RV). For each one, we use a publicly available whole heart segmentation dataset Zhuang et al. (2019) featuring 120 3D whole-heart models to learn a SDF auto-decoder model Park et al. (2019a). We reserve 20 samples for validation and use the rest for training the latent implicit models. The contact ratios between these components are pre-computed based on the training data and are used to formulate the constraints during reconstruction. The nn-Unet segmentation model is trained on 15 cardiac images of the same dataset. Note that only images of 20 cases are publicly available.

We test our method on two separate datasets: First a public test-set of 5 cardiac images from the same distribution as the training data but that is *not* part of it, and second an in-house dataset consisting of 10 cardiac images obtained from a nearby hospital's radiology department. The latter serves as an out-of-distribution test set due to significant domain gaps between the training and testing images, particularly in terms of image quality. For each test CT image, we first run a nn-Unet Isensee et al. (2018) trained on 15 fully labeled heart CT images from the training set. We then fit an SDF to each heart component and refine them jointly while imposing the constraints of Sec. 3.2. We measure topological correctness via the average surface penetration (%) and the average absolute difference contact ratio (%) from the ground-truth.

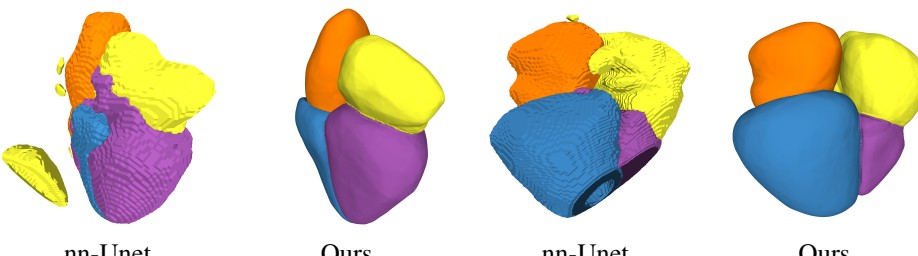

|  nn-Unet | Ours | nn-Unet | Ours |

Figure 4: **Heart Reconstruction from two Out-of-Distribution samples.** The nn-Unet result contains many mistakes, which our approach fixes.

Fig. 3 depicts a qualitative result on the public dataset Zhuang et al. (2019) where the test examples are in-distribution (ID) with respect to the training data, while Fig. 4 depicts qualitative results on the private dataset where the test examples are essentially out-of-distribution (OOD). Tab. 1 summarizes our quantitative results on the two datasets (also see Tab. 5 for the raw contact ratio values). Unsurprisingly, in the ID case nn-Unet does well with proper topological connectivity and low reconstruction errors. Nevertheless, fitting SDFs independently without constraints yields topological errors. This is because even small small misalignments between object boundaries can result in interpenetration. Our method fits all SDFs jointly and enforces the constraints, resulting in accurate composite objects with smooth surfaces and proper topology, such as ones in Fig.3b. For the OOD samples, nn-Unet results are highly inaccurate with misaligned boundaries and various isolated parts, as can be seen in Fig.4. When shape reconstruction error is significantly high, topological correctness becomes irrelevant. Fitting an SDF to the individual parts greatly reduces the Chamfer distance error by a factor of more than 10 because the latent vector model imposes priors on the noisy data. However, the results exhibit severe topological errors. In contrast, fitting SDFs jointly and enforcing the constraints further reduces the Chamfer distance error and only with minimal topological mistakes, which is due to the instability when fitting SDFs on extremely noisy input shapes.

Table 1: **Heart reconstruction.** We report the average surface penetration P.(%) (lower is better), the average absolute difference from the ground-truth contact ratio $|\Delta CT|$ (lower is better), and Chamfer distance. Shapes generated using the baseline tend to intersect each other while having lower contact ratios than those obtained by optimizing all parts jointly and imposing the constraints. Shapes from nn-Unet exhibit correct topology but are extremely inaccurate for OOD samples.

| Method | LV - MLV | | LV-LA | | MLV-RV | | All |
|---|---|---|---|---|---|---|---|
| | P.(%) | $|\Delta CT|$ (%) | P.(%) | $|\Delta CT|$ (%) | P.(%) | $|\Delta CT|$ (%) | CD ($\times 10^3$) |
| | | | In-distribution samples from public test set | | | | |
| nn-Unet | 0.0 | 0.1 | 0.0 | 0.1 | 0.0 | 0.1 | 0.3 |
| Individual | 11.1 | 11.7 | 1.7 | 2.7 | 3.2 | 3.6 | 0.4 |
| Joint (Ours) | 0.0 | 0.1 | 0.0 | 0.1 | 0.0 | 0.0 | **0.3** |
| | | | Out-of-distribution samples from private test set | | | | |
| nn-Unet | 0.0 | 0.8 | 0.0 | 1.9 | 0.0 | 0.3 | 46.8 |
| Individual | 11.2 | 14.7 | 1.7 | 2.7 | 2.8 | 3.6 | 3.6 |
| Joint (Ours) | 0.2 | 4.4 | 0.3 | 0.3 | 0.4 | 1.0 | **3.1** |

**3D Lumbar Spine Segmentation.** We also use our method to reconstruct vertebrae in a dataset containing 460 CT images of the five lower vertebrae of the human spine (L1-L5) Wasserthal et al. (2022). Of these, 80% are used to train both the nn-UNet and the latent implicit models, while 10% samples are reserved for testing and 10% are for validation. In the entire dataset, each pair of adjacent vertebrae exhibits a minimum gap of 1 pixel, which is the constraint we enforce during reconstruction. In Tab. 2, we report topological errors measured by the number of contact vertices (N) and the areas of contact surfaces (px$^2$), along with reconstruction accuracy in terms of Chamfer distances. Our approach yields reconstructions with almost no constraint violations and achieves the lowest reconstruction errors. Note that enforcing the minimum gap constraint is particularly challenging because there are only small volumes surrounding each vertebrae joints that are topo-

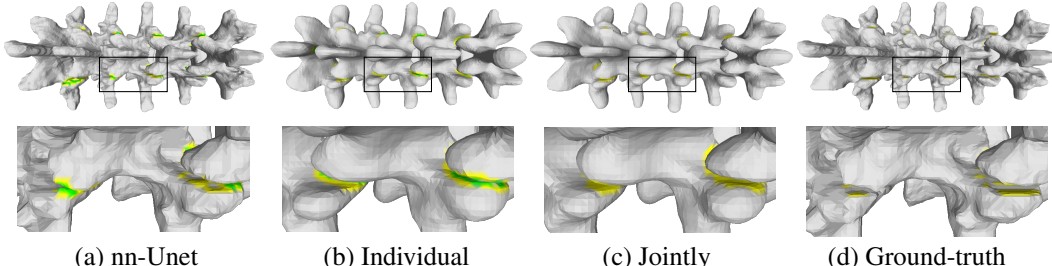

|              |                |             |                  |
| :----------: | :------------: | :---------: | :--------------: |
| (a) nn-Unet  | (b) Individual | (c) Jointly | (d) Ground-truth |

Figure 5: **Verterbrae reconstruction.** Second row shows zoom-in crops of the area marked in the first row. Yellow denotes close proximity to another shape, which is allowable when there is a remaining gap. Green indicates actual contact, which should not occur. (a) nn-Unet segmentation with many unwanted green areas. (b) Fitting SDFs individually to each component. There are still with many green areas. (c) Fitting SDFs jointly and imposing constraints using our method. The green areas have disappeared and have replaced with yellow areas denoting acceptable proximity. (d) Ground-truth with similar yellow areas.

logically relevant, as can be seen in the highlighted areas in Fig. 14. Our loss functions directly modify these critical areas to yield constraint-compliance shapes.

Table 2: **Spine reconstruction.** We compare three methods based on surface area and the number of points violating the minimum distance constraint (lower is better), as well as Chamfer distance (CD, lower is better). The results are presented for different vertebrae pairs (L1-L2, L2-L3, L3-L4, L4-L5) and averaged across all shapes. Our method shows almost no constraint violations and a smaller Chamfer distance, indicating superior reconstruction accuracy compared to nn-Unet and DeepSDF.

| Method | L1 - L2 | | L2 - L3 | | L3 - L4 | | L4 - L5 | | All |
| --- | --- | --- | --- | --- | --- | --- | --- | --- | --- |
| | $px^2$ | N | $px^2$ | N | $px^2$ | N | $px^2$ | N | CD ($\times 10^3$) |
| nn-Unet | 261.4 | 123.2 | 527.2 | 237.3 | 644.6 | 294.3 | 761.1 | 358.9 | 0.4 |
| Individual | 315.8 | 154.2 | 309.8 | 166.1 | 416.4 | 227.7 | 503.3 | 279.4 | 0.4 |
| Joint (Ours) | **0.0** | **0.8** | **0.5** | **2.3** | **0.5** | **2.9** | **1.2** | **3.9** | **0.3** |

**Parallel Surface Reconstruction.** Our approach can also be used to model parallel surfaces, such as skin layers of the same organ, as shown in Fig. 6. To show this, we start from the mesh of each heart component and apply a 3D erosion operation to obtain an inner structure of each part. We consider the original structure as the "outer layer" and the eroded one as the "inner layer", which are colored blue and yellow respectively in Fig. 6. We use our proposed method to simultaneously model these two structures. In this experiment, they maintain a minimum distance of 2 pixels and a maximum distance of 4 pixels, which are the constraints we want to enforce. To do so, we extend our framework to enforce both a minimum distance and a maximum distance constraint, similar to Eq. 11. For measuring topological correctness, we count the number of vertices of the inner object surface that violate the constraint. The results are given in Tab. 4 (Appendix), showing that modeling parts individually yields shapes with 2-3% violated vertex, while our proposed method exhibits minimal violations.

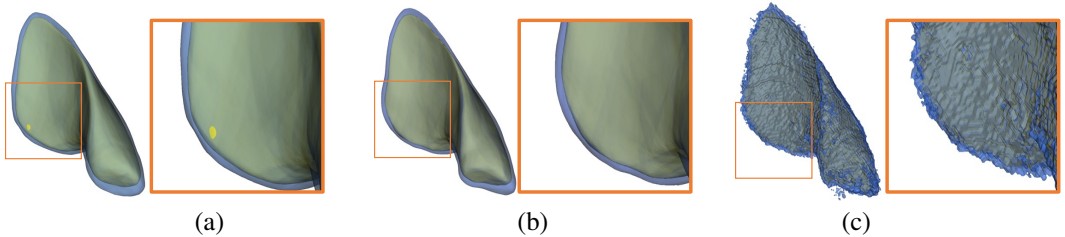

|     |     |     |
| :-: | :-: | :-: |
| (a) | (b) | (c) |

Figure 6: **Reconstructing thin layers.** In each case, the image to the right is a magnified version of part of the image on the left. (a) Fitting two layers separately. Note the yellow dot that denotes the inner surface penetrating the outer one. (b) Fitting the layers jointly while imposing the parallel constraints. (c) Ground truth with voxels belonging to different layers painted in different colors.

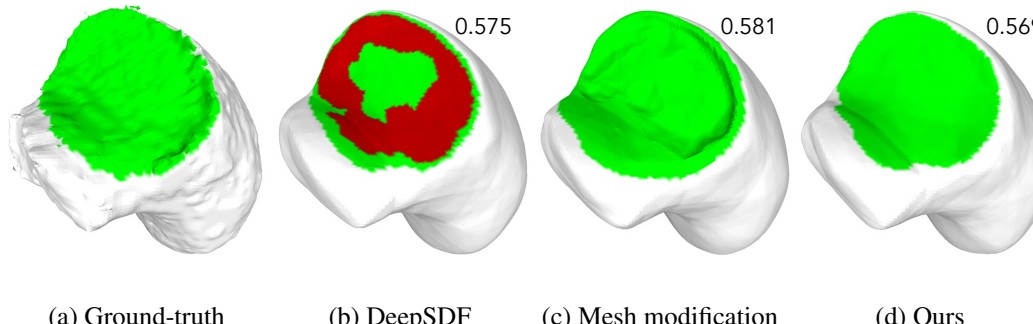

| (a) Ground-truth | (b) DeepSDF | (c) Mesh modification | (d) Ours |

Figure 7: **Comparison with explicit mesh modification.** Starting from meshes obtained via DeepSDF, we select violated vertices and move them along their surface normal to resolve object-penetration. This approach yields rough surfaces with increased reconstruction errors. The numbers on the top-right corner indicates chamfer distances ($\times 10e4$).

## 4.1 ABLATION STUDY

Table 3: **Ablation study.** Performance of our method when reconstructing a left-ventricle and its myocardium when removing specific loss components.

| Method | w/o $L_{inter}$ | w/o $L_{contact}$ | w/o $L_{non-contact}$ | w/o $L_{data}$ | All | GT |
|---|---|---|---|---|---|---|
| Penetration | 5% | 0% | 0% | 0% | 0% | 0% |
| Contact | 17.2% | 13% | 29% | 26% | 27% | 27% |
| CD ($\times 10^3$) | 3.8 | 3.7 | 3.5 | 33.2 | 3.1 | 0 |

We verify the effect of each loss function in our framework by conducting ablation studies where we omit each of the loss term and measure both topological error and reconstruction errors when reconstructing a pair of left-ventricle and its myocardium in the out-of-distribution testing set. As can be seen in Tab. 3, a joint system of all losses is essential for reconstructing constraint-compliance objects.

An alternative approach to refining the contact ratio between two surfaces is modifying their explicit meshes. Specifically, we first fit DeepSDF to the segmentation output of nn-Unet and then convert them into explicit meshes. To ensure these meshes contact each other with a contact ratio k%, we identify the top k% vertices with the smallest summed distances to both objects, and subsequently adjusting their positions such that those vertices become contacting points, *i.e.*, lying on the surfaces of both objects. To adjust the positions of those points, we move them along the surface normals. However, this approach yields rough surfaces and a noticeable increase in reconstruction errors, as can be seen in Fig. 7. This is because the movements along surface normals often lead to unrealistic deformations and it is challenging to maintain the shape integrity and the geometric continuity when doing so. Further, this approach is computationally demanding due to the additional steps required for vertex selection and adjustment, especially when there are more than just two objects to be refined.

## 5 CONCLUSIONS

We have introduced a novel method to incorporate topological constraints into 3D multiple-object reconstructions. In the heart reconstruction case, our method enforces implicit objects to maintain a precise contact ratio while preventing penetration as in various ventricles must come into contact to facilitate blood flow without overlapping. In the lumbar spine reconstruction case, we ensure that each pair of adjacent vertebrae does not contact each other. We demonstrate that the topological relationships between implicit objects can be effectively observed and adjusted through uniform sampling. Future work will focus on more complex scenarios, such as enforcing specific areas of contact or managing interactions among dynamic objects. Additionally, a mechanism to not only enforce but also detect when a constraint cannot be satisfied for a given image could serve as a powerful diagnostic tool. Finally, supervision from constraint-aware implicit models could potentially guide image segmentation models to focus on topologically meaningful areas.

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

## A   MORE DETAILS ON NETWORK ARCHITECTURE AND OPTIMIZATION

In all of our experiments, we adopt the auto-decoder framework introduced in deepSDF Park et al. (2019b). The decoder network consists of 6 fully connected layers, each with 256 dimensions and ReLU activation. We utilize a Tanh activation function as the final layer to output the signed distance function (SDF) values. Layer normalization Ba et al. (2016) is applied to normalize intermediate outputs after each layer. The auto-decoder is trained for 20,000 epochs using the Adam optimizer Kingma & Ba (2014).

To prepare data, we normalize the vertices of each mesh to the range of [-1,1]. For training, we sample 440,000 spatial points to compute SDF values for each shape. The ratio of points near the object's surface to random points is set to 10:1. The baseline deepSDF models are trained for 20,000 epochs. Throughout the optimization process, we sample 220,000 spatial points to compute SDF values for each shape.

To refine the topological interaction, we uniformly sample 300000 random points after each 10 training iterations. Computing SDF distances from these points to deepSDF models requires a single feed-forward pass through each model and obtaining anchor points takes 0.15 seconds in total. For each testing instance, we optimize the system for 2000 iterations, which takes 28 seconds for an NVIDIA A100 GPU. The baseline deepSDF takes 22 seconds. We set all our hyper-parameters via the performance of the model on validation data. In the heart reconstruction case, $(\lambda_1, \lambda_2, \lambda_3, \lambda_4)$ are set to $(1, 5, 1, 10)$ while in the spine reconstruction case, $(\lambda_1, \lambda_2)$ are set to $(1, 10)$.

## B   TOPOLOGICALLY MEANINGFUL POINTS VISUALIZATION

We visualize anchor points used in our method for refining interactions between different pairs of heart components. We choose among randomly sampled points ones that are close to the surfaces of both objects where the numbers of points are determined via a prior contact ratio. As can be seen, those anchor points are distributed evenly on the contact surface, allowing us to refine the interactions between the two objects.

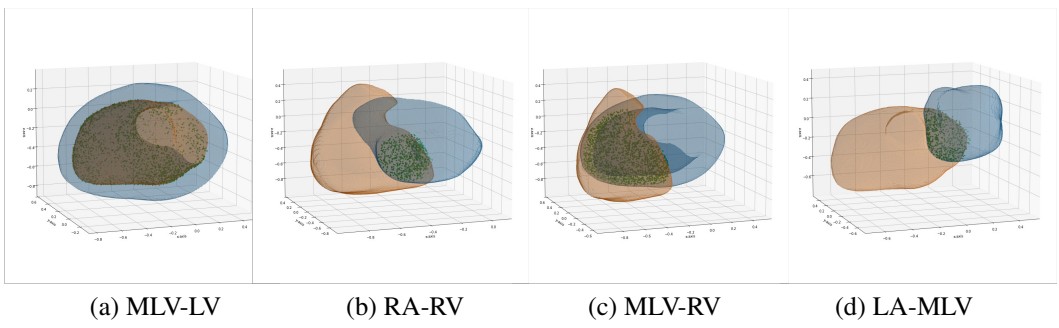

|                |                |                |                |
| :------------: | :------------: | :------------: | :------------: |
|  (a) MLV-LV    |   (b) RA-RV    |   (c) MLV-RV   |  (d) LA-MLV    |

Figure 8: **Topologically Meaningful Points Visualization.** In heart reconstruction case, topologically meaningful points typically reside close to both objects.

## C   DOMAIN GAP BETWEEN THE PUBLIC TRAINING DATA AND THE PRIVATE TEST SET.

There is a pronounced domain gap between the in-house data and the public dataset, driven by distinct imaging conditions, as shown in Fig. 9. Images in the private test set have lower resolution and blurred edges, presenting a critical challenge for our model's ability to transfer and adapt to real-world clinical environments. Due to the limited training data and the absence of such challenging cases, nn-Unet fails to produce satisfactory results on this private test set.

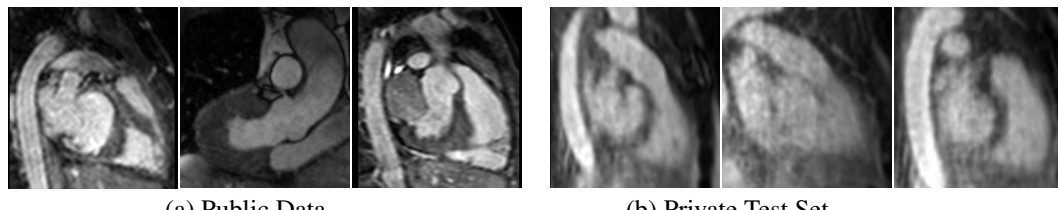

(a) Public Data            (b) Private Test Set

Figure 9: **Domain Gap between the public data Zhuang et al. (2019) and In-House Data.**

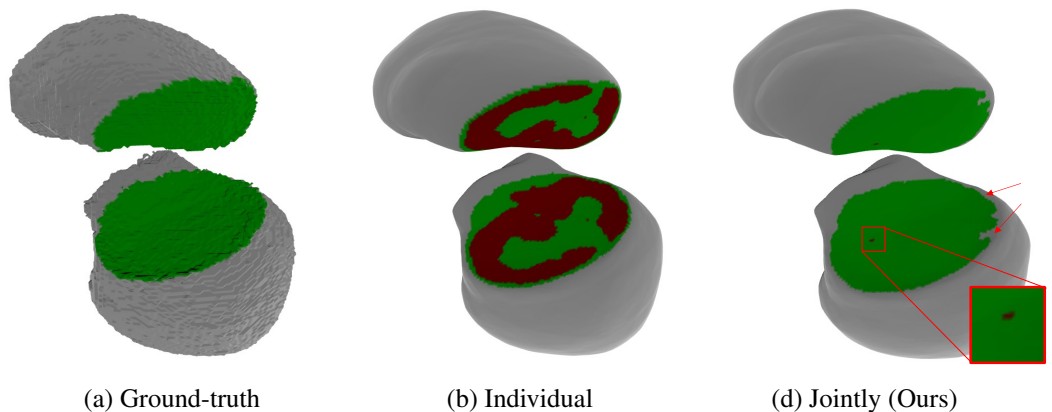

(a) Ground-truth       (b) Individual       (d) Jointly (Ours)

Figure 10: **A failure case of our method.** Our method can resolve most of the topologically incorrect areas. However, there may remain small subsets of points that violate the constraints while being challenging to identify.

## D    FAILURE CASES

Our method can resolve most of the topologically incorrect areas. However, there may remain small subsets of points that violate the constraints while being challenging to identify. Our existing approach hinges on uniformly distributed sample points, which inherently lacks the precision required for localizing small erroneous areas. A potentially fruitful avenue for future exploration involves identifying regions particularly susceptible to topological errors and concentrating our efforts there, perhaps via an adaptive sampling scheme. Furthermore, it is also difficult to constrain two shapes to contact at the non-smooth surface areas (sharp edges), due to the smooth representation inherent to implicit functions.

Table 4: **Reconstructing outer-inner objects.** The inner object surface is expected to be in a distance of 2-4 pixels from the outer object surface.

| Component | Violation (N.o. Vertices) | | Total Vertices |
|---|---|---|---|
| | deepSDF | Ours | |
| Myocardium-LV | 4638.94 | 145.80 | 132104.5 |
| Left-Atrium | 1170.26 | 0.00 | 43932.0 |
| Left-Ventricle | 2020.36 | 0.00 | 60126.6 |
| Right-Atrium | 1267.76 | 55.66 | 36288.4 |
| Right-Ventricle | 1547.81 | 16.59 | 67140.2 |

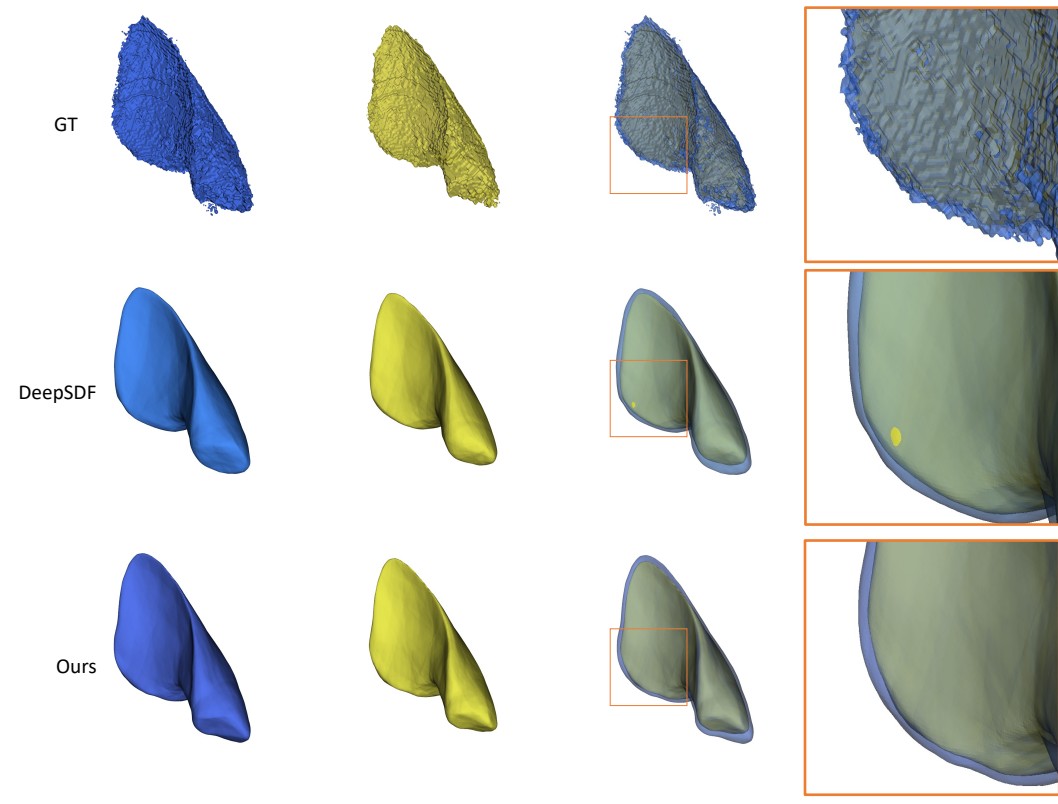

|           Outer           |           Inner           |        Outer&Inner        |        Zoom-in Crop       |

Figure 11: **Reconstructing a pair of outer-inner objects.** Both the baseline method DeepSDF and our method can accurately reconstruct the overall shapes of objects. The last column highlights an area where a partial surface of the inner object reconstructed by DeepSDF is outside the outer object. There is no violation in our case.

## E    PARALLEL SURFACE RECONSTRUCTION

Tab.4 compares the number of vertices violating the expected 2-4 pixel distance between the inner and outer object surfaces for deepSDF and our method. Our approach significantly reduces violations across all components, achieving near perfect compliance in most cases. Fig.11 compares 3D contour reconstructions of outer and inner objects using the baseline method deepSDF and our method. Both approaches successfully capture the overall object shapes. However, the zoom-in crop in the last column reveals a violation in the deepSDF reconstruction, where part of the inner object surface extends outside the outer object, a mistake that is not present in our method. We show some 2D slices to further demonstrate this in Fig.12. The baseline deepSDF model fails to maintain this distance in several cases, as highlighted by the red circles. In contrast, our method consistently satisfies the constraint across all examples.

## F    ADDITIONAL VISUALIZATION AND TABLE

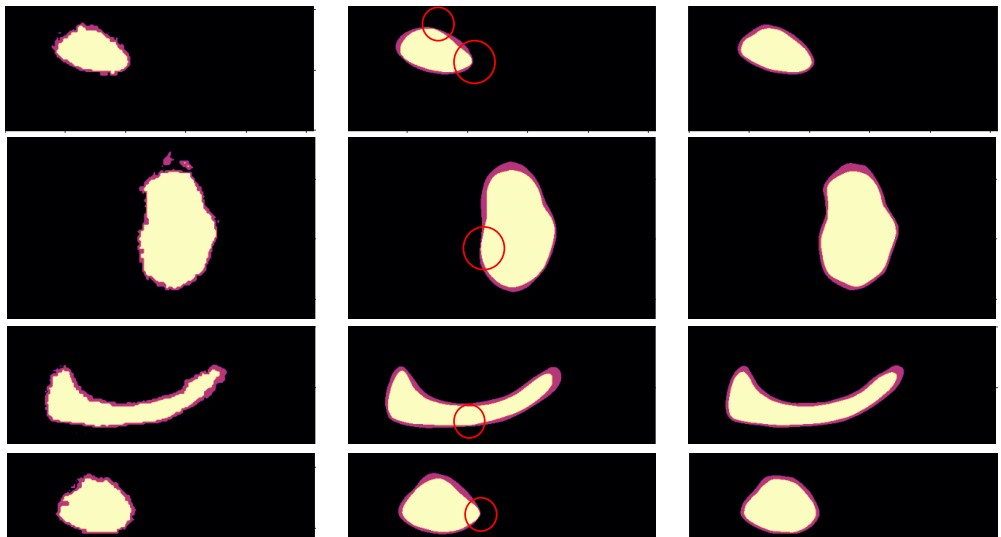

Figure 12: **Reconstructing a pair of outer-inner objects.** We aim to reconstruct a pair of outer-inner objects where the distance between them is in a fixed range. We visualize several cases where a baseline DeepSDF model fails to generate objects that satisfy this constraint (circled in red). Our method works well in all cases.

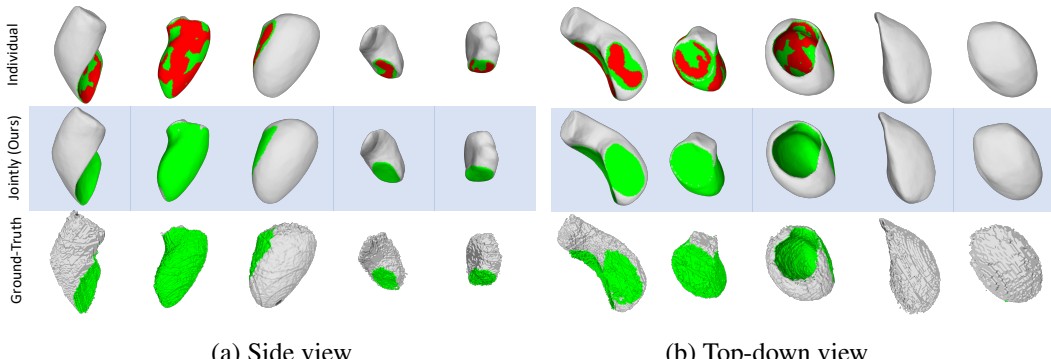

(a) Side view          (b) Top-down view

Figure 13: **Different Human Heart Components.** We show the reconstructed heart components from the baseline in the top row and our method in the middle row. The ground-truth is shown the bottom row. Green depicts proper contact while red means penetration.

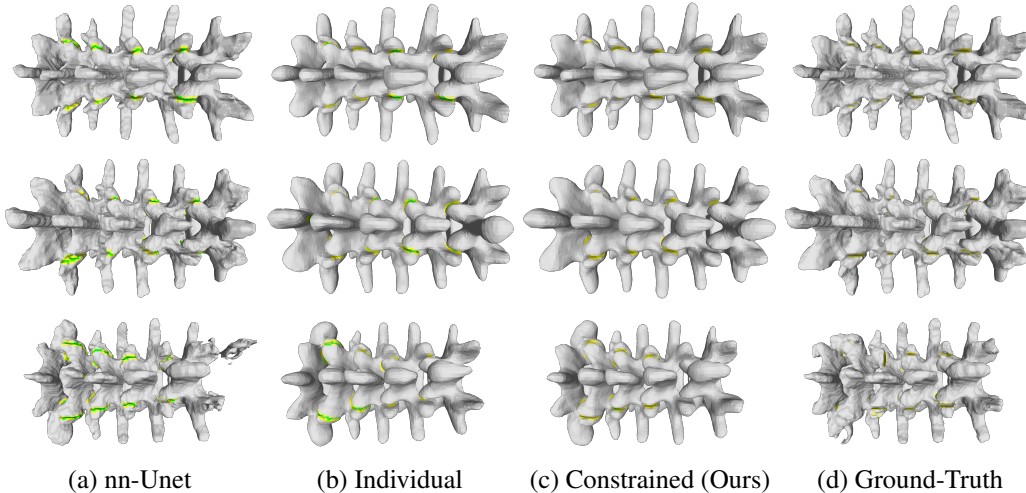

(a) nn-Unet     (b) Individual     (c) Constrained (Ours)     (d) Ground-Truth

Figure 14: **3D Verterbea refinement via Implicit Functions.** (a) shows the segmentation results from nn-Unet. (b) depicts the resulting shapes when fitting implicit functions to each individual nn-Unet segmentation part, and (c) shows the results when fitting all parts simultaneously, with constraints enforced via our method. Areas in yellow indicate points in close proximity to other shapes, while areas in green highlight instances of touching, which should not occur.

Table 5: **Heart reconstruction.** We report the average surface penetration (%) (lower is better), contact ratio (%) (closer to the ground truth is better), and chamfer distance (lower is better). Shapes generated from a vanilla deep-SDF model tends to intersect each other while having lower contact ratios.

| Method | LV - MLV | | LV-LA | | MLV-RV | | All |
|---|---|---|---|---|---|---|---|
| | Pen.(%) | Cont.(%) | Pen.(%) | Cont.(%) | Pen.(%) | Cont.(%) | CD ($\times 10^3$) |
| | In-distribution public test set | | | | | | |
| nn-Unet | - | 26.9 | - | 4.9 | - | 4.8 | 0.3 |
| deepSDF | 11.1 | 15.3 | 1.7 | 2.3 | 3.2 | 1.1 | 0.4 |
| Ours | 0.0 | 27.1 | 0.0 | 4.9 | 0.0 | 4.7 | **0.3** |
| | Out-of-distribution private test set | | | | | | |
| nn-Unet | - | 26.2 | - | 6.9 | - | 5.0 | 46.8 |
| deepSDF | 11.2 | 12.3 | 1.7 | 2.3 | 2.8 | 1.1 | 3.6 |
| Ours | 0.2 | 22.6 | 0.3 | 4.7 | 0.4 | 3.7 | **3.1** |
| GT | - | 27.0 | - | 5.0 | - | 4.7 | - |

