# OpenReview forum: "Enforcing 3D Topological Constraints in Composite Objects via Implicit Functions"
_ICLR.cc/2025/Conference — ICLR 2025 Conference Withdrawn Submission_

### Official Review · Reviewer_9fUj · 2024-10-18

**Soundness:** 2
**Presentation:** 2
**Contribution:** 1
**Rating:** 3
**Confidence:** 5

**Summary:**

## Motivation
- Deep implicit functions have emerged as a powerful solution for representing 3D shapes.
- However, most of the focus has been put on single-object scenarios, ignoring topological constraints (contact enforcement, non-interpenetration, etc.) that may arise in multi-object applications, such as anatomical modeling.

## Contributions
- The authors extends existing neural SDF solutions [Park et al., 2019] to enforce non-interpenetration between different object categories (i.e., different anatomical entities), as well as to enforce user-defined surface contact ratio or surface distance.
- This is achieved through the introduction of attraction-repulsion losses applied to a subset of 3D points meeting the contact constraints.

## Results
- The authors demonstrate their solution on two clinical use-cases: 3D whole-heart reconstruction (enforcing user-defined surface contact ratio between hear components) and lumbar spine reconstruction (enforcing user-defined minimal distance between vertabrae).
- They compare to the original segmentation results (nn-Unet [Isensee et al., 2018]) as well as baseline DeepSDF [Park et al., 2019a], showing that their method succeeds in enforcing non-interpenetration and the user-defined constraints.
- An ablation study, as well as well-presented qualitative results, are also provided.

**Strengths:**

_(somewhat ordered from most to least important)_

## S1. Clear Illustrations
- The authors provide well-designed illustrations to convey their intuition/contributions (Fig. 2), as well as to share their qualitative results (e.g., by highlighting contact vs. interpenetration regions in Fig. 3).

## S2. Motivation & Relevance
- The implicit modeling of multi-component scenes is an under-explored topic. Most of the research in that direction focuses on human/object interaction scenarios, but the resulting solutions do not always transfer well to anatomical use-cases (e.g., due to rigidity assumptions).
- Moreover, the authors' idea to condition the contact/distance losses based on medical prior is interesting and well-motivated.

## S3. Decent Reproducibility
- Even though the authors did not release their code, an expert in the art should be able to re-implement their work, i.e., extending the publicly-available DeepSDF implementation with the proposed multi-object losses.

**Weaknesses:**

_(somewhat ordered from most to least important)_

## W1. Lack of Relevant SOTA Comparison

### W1.a. No Mention of Existing Multi-Organ DIF Works

[L154-157] The authors claim that:

> We focus on two different kinds of constraints—**neither of which has been
considered in previous work**—in two distinct scenarios. First, when reconstructing the four chambers
of the human heart, these chambers **should never intersect but instead should be in contact with
each other** over a given percentage of their surface areas. [...]

However, their novelty claim is heavily questionable. Even when focusing only on the narrow domain of implicit anatomical modeling, at least two papers [a, b] have already proposed contact and/or non-interpenetration losses. Similar losses have been proposed for other applications, e.g., human/object interaction modeling [c]. The fact that the authors neither compare to—nor even discuss—such prior art is problematic.

### W1.b. Comparison to Baseline Only
Similarly, the authors only compare their method to a single other deep implicit function method, DeepSDF [Park et al., 2019]. This work is quite outdated and focuses on single-object scenarios. It is obvious that it would under-performed the proposed solution w.r.t. contact/interpenetration metrics. It would have been meaningful to compare the proposed method to (a) more recent implicit solutions targeting multi-object scenarios [a,b,c] ; or at least to (b) DeepSDF applied to modeling the entire scene (as one single multi-part object) rather than to multiple DeepSDF instances applied to each component.

## W2. Superficial Contributions Compared to SOTA

### W2.a. Attraction-Repulsion Losses Already Applied to Anatomical Modeling

With the above-mentioned prior art in mind, the contributions claimed in this paper appear rather shallow. Their only claims are the losses ensuring non-interpenetration, as well as enforcing surface contact or surface distance (depending on the scenario). While the idea to condition the contact/distance losses on user-defined values is novel, similar contact/repulsion functions already exist in the literature [a,b,c]. Due to the lack of comparison, it is also unclear how their formulation of the contact/inter-penetration losses fair compared to existing solutions.

### W2.b Redundant Definition (?)

The self-intersection loss $\mathcal{L}\_{\text{intersecting}}$ and contact-ratio loss $\mathcal{L}\_{\text{contact}}$ proposed in this paper appears somewhat redundant, as well as highly similar to the loss $\mathcal{L}^\mathcal{C}$ proposed in [b], where it is defined as an "attraction-repulsion" function to ensure both non-interpenetration and contact of surfaces.

Similar to the current submission, the loss in [b] relies on the sampling of contact points (set $\mathcal{C}$ in [b]), generalized to any number of surfaces (not just 2). The only contribution of the present paper is the weighting of the set size by the target user-provided contact ratio (a minor change, in my opinion).

Indeed, if we define:

$\mathcal{A}\_{\text{contact}} = \mathcal{A}\_{\text{intersecting}} \cup \mathcal{A}\_{\text{outside}} \cup \mathcal{A}\_{\text{single}}$,

with $\mathcal{A}\_{\text{outside}}$ set of close points outside all objects and $\mathcal{A}\_{\text{single}}$ set of points inside a single object, then:

$\mathcal{L}\_{\text{contact}} = \sum\_{x \in \mathcal{A}\_{\text{contact}}} |\sum\_{i \in [a, b]} f(i, x)| $
$  = \sum\_{x \in \mathcal{A}\_{\text{intersecting}}} |\sum\_{i \in [a, b]} f(i, x)| + \sum\_{x \in \mathcal{A}\_{\text{outside}}} |\sum\_{i \in [a, b]} f(i, x)| + \sum\_{x \in \mathcal{A}\_{\text{single}}} |\sum\_{i \in [a, b]} f(i, x)|$
$  = \sum\_{x \in \mathcal{A}\_{\text{intersecting}}} \sum\_{i \in [a, b]} |f(i, x)| + \sum\_{x \in \mathcal{A}\_{\text{outside}}} \sum\_{i \in [a, b]} |f(i, x)| + \sum\_{x \in \mathcal{A}\_{\text{single}}} |\sum\_{i \in [a, b]} f(i, x)|$
$  = \mathcal{L}\_{\text{intersecting}} + \sum\_{x \in \mathcal{A}\_{\text{outside}}} \sum\_{i \in [a, b]} |f(i, x)| + \sum\_{x \in \mathcal{A}\_{\text{single}}} |\sum\_{i \in [a, b]} f(i, x)|$,

c.f. $| x + y | = | x | + | y |$ if $\text{sign}(x) =  \text{sign}(y)$

Hence $ \mathcal{L}\_{\text{intersecting}}$ being redundant to  $\mathcal{L}\_{\text{contact}}$.

Moreover, based on the above equation, we can also observe that:

$\mathcal{L}\_{\text{contact}} \approx \mathcal{L}^\mathcal{C} + \Delta\mathcal{L}$,

with the main difference (if we ignore the sigmoid-based normalization added to the loss $\mathcal{L}^\mathcal{C}$ in [b]) being:

$\Delta\mathcal{L} = \sum\_{x \in \mathcal{A}\_{\text{single}}} |\sum\_{i \in [a, b]} f(i, x)| - \sum\_{x \in \mathcal{A}\_{\text{single}}} \sum\_{i \in [a, b]} |f(i, x)|$.

I.e., for points close to 2 objects but inside only one, the authors of [b] compute the sum of absolute SDF values, whereas the present authors compute the absolute sum of SDF values. I do not have the insight to know which is best (a comparison could be interesting), but I believe that the difference in terms of overall supervision is minor (since it concerns only a small subset of points, and since other losses such as $\mathcal{L}\_{\text{data}}$ would have a more significant influence on those).

## W3. Medical Grounding & Clinical Applicability
- A key claim in this work is the enforcement of topological priors from the medical literature. However, the medical grounding is somewhat lacking. E.g., it is unclear where the authors got the 27\% value used as surface contact ratio for left ventricle and left myocardium. Only one reference is provided w.r.t. heart anatomy [Buckberg et al., 2018], but the above number does not seem to actually appear in that referenced article (?).
- One can also wonder what would be the actual clinical use for a  method that forces the reconstruction to meet statistical constraints based on healthy populations. E.g., what happens for patient with a heart or spine condition? The authors do warn that "_in this paper, we restrict ourselves to healthy subjects for whom this constraint must be satisfied._" [L177-178] But they do not provide any insight on the clinical impact of this limitation.

## W4. Minor - Methodology Not Always Clear
- The contributions w.r.t. enforcing the contact ratio and w.r.t. enforcing the minimum distance appear severely disconnected (both in terms of methodology and in terms of actual application). The formalism of the corresponding losses could be better homogenized, e.g., by highlighting how the two losses constrain the range of valid distances (the contact loss enforce a maximum distance ; the distance loss enforce a minimum one).
- The redundant definition of the point sets ($\mathcal{A}\_{\text{contact}}, \mathcal{A}\_{\text{non-contact}}, \mathcal{A}\_{\text{intersecting}}$) is a bit confusing. I.e., is it useful to list these sets in [L221-223] if they are formally defined afterwards, [L255-258]?
- The font style of the loss functions is not always consistent (.e.g, $\mathcal{L}\_{\text{contact}}$ vs. $\mathcal{L}\_{contact}$).

#### **Additional References:**

[a] Zhang, Congyi, et al. "An Implicit Parametric Morphable Dental Model." ACM Transactions on Graphics (TOG) 41.6 (2022): 1-13.

[b] Liu, Yuchun, et al. "Implicit Modeling of Non-rigid Objects with Cross-Category Signals." Proceedings of the AAAI Conference on Artificial Intelligence. Vol. 38. No. 4. 2024.

[c] Hassan, Mohamed, et al. "Synthesizing physical character-scene interactions." ACM SIGGRAPH 2023 Conference Proceedings. 2023.

**Questions:**

_see **Weaknesses** for key questions/remarks._

---

### Official Review · Reviewer_wEaN · 2024-10-28

**Soundness:** 2
**Presentation:** 4
**Contribution:** 2
**Rating:** 3
**Confidence:** 5

**Summary:**

The paper introduces a concept to incorporate topological constraints for cardiac shape representation in the context of deep signed distance functions. The method is composed of several parts: sampling of topologically meaningful points, optimization of the sum of four loss functions, and the enforcement of minimum distance constraints. In several numerical experiments, the performance of the method is both qualitatively and quantitatively examined. In particular, an ablation study reveals that all four loss functions are essential to achieve the reported performance.

**Strengths:**

The presentation of the paper is excellent, and all methods are clearly described. To the best of my knowledge, I have not seen the combination of the four loss functions in this way (although, I have encountered most (probably all) as separate loss functions elsewhere).
Moreover, the research question itself (imposing topological constraints for DeepSDF) is highly significant.
Finally, the numerical experiments are systematically conducted and (partially) underline the claims of the paper.

**Weaknesses:**

The reconstruction method itself builds upon rather old publications by Park and Isensee, thereby completely ignoring the regularized deepSDF approaches with their substantially improved reconstruction quality (e.g., "Reconstruction and completion of high-resolution 3D cardiac shapes using anisotropic CMRI segmentations and continuous implicit neural representations." by Sander et al., "Sdf4chd: Generative modeling of cardiac anatomies with congenital heart defects." by Kong et al. or “Shape of my heart: Cardiac models through learned signed distance functions” by Verhülsdonk et al.). In particular, these regularized versions are proven to preserve topological constraints better. A systematic benchmark with some of these recent approaches is required instead of only considering "old" approaches.
Moreover, the design of the four loss functions is entirely heuristic, any motivation or mathematical reasoning for this particular choice is completely lacking (only the ablation study partially underlines this specific choice).
Finally, I suspect that the sampling requirements (300k points after 10 iterations) result in inferior run time (and maybe performance) compared to the above-mentioned regularized approaches.

**Questions:**

1. Have you integrated this method into regularized versions of the deep signed distance (see section weaknesses)? Can you please report on the results? Here, the integration of a Lipschitz regularization would be one possible option.
2. Can you provide a justification for the inclusion of these four particular loss functions beyond the ablation study? Are there particular theoretical frameworks or principles that should be applied to justify the loss function design?
3. What is the additional runtime caused by the sampling?
4. I am lacking details on the optimization in Section 3.2.3. Can you please provide them?
5. In Figure 8, I can hardly recognize the distribution of the topologically meaningful points near the interfaces. Can you please present this in a better way?

---

### Official Review · Reviewer_6aXe · 2024-11-04

**Soundness:** 3
**Presentation:** 3
**Contribution:** 3
**Rating:** 3
**Confidence:** 5

**Summary:**

This work focuses on resolving the shape contacting issue using the optimization in post-processing. Two constraints are proposed to regularize the shape representation: contact ratio and minimum distance between two shapes. By keeping the desired contact ratio and keeping the distance between shapes, the reconstructed 3D shape would be more precise with less penetration artifacts.

**Strengths:**

1. This work targets on exploring the shape constraints for reconstruction the organs for the human scan.
2. In this paper, the authors propose two shape constraints, one is contact ratio and another is the minimum distance. Several straightforward losses are introduced to keep the desired contact ratio and distance by optimization.
3. The writing is clear and easy to follow.

**Weaknesses:**

1. Utilizing the segmentation from the existing models and DeepSDF to fit the segmentation, the proposed method is specifically designed for the shape post-processing with knowledge from the previous steps.
2. The P_contact and P_non-contact are from the overfitted DeepSDF representation. However, if the DeepSDF representation is not correct or the segmentation is not accurate enough, the P_contact points set are not correct. The optimization result therefore can not adjust the initial prediction and optimization result will have artifact. Please discuss how the method handles cases where the initial DeepSDF representation or segmentation is inaccurate. An analysis of the method's robustness to errors in the initial inputs would benefit.
3. In the introduced loss function, the optimization only applied on the 3d shape representation, however, the optimized 3D shape might not consistent with the image after the optimization. Combined with last point, if the initial representation or segmentation is inaccurate, how the optimization could adjust the errors. Please include a discussion on potential methods to maintain this consistency or evaluate it quantitatively.

**Questions:**

Please refer to the weakness part.
Additionally, the abdomen dataset should be a perfect fit for this work as abdomen region contains multiple organs and they are close to each other.

---

### Official Review · Reviewer_hb8a · 2024-11-04

**Soundness:** 3
**Presentation:** 4
**Contribution:** 3
**Rating:** 5
**Confidence:** 4

**Summary:**

The authors propose a method for 3D organ reconstruction with regard to pre-defined topological constraints. The core of the proposed approach is a global Monte Carlo sampling that evaluates the relationship between signed distances for two organs to estimate their relationship. In contrast, previous works only consider local constraints, e.g. non-intersection of different parts, but cannot evaluate the global contract ratio of two sub-organs. The authors evaluate their method on both multi-organ cardiac and spine datasets, but emphasize applicability to other organs.

**Strengths:**

- S1: The work is interesting and approaches a worthwhile topic in the subdomain of medical image analysis. The authors aptly note that existing multi-organ segmentation methods do not consider topological constraints between different sets of organs. While the surface can be reconstructed to avoid local artefacts, constraints between organs with specific priors cannot be easily specified. Based on this, they propose several loss functions based on surface-aware Monte Carlo sampling that propose a correct behaviour (contact, non-contact, non-intersection) between two shape pairs.
- S2: The presentation is compelling and professional; there are no major typos, and the figures are nicely constructed.
- S3: The method works well with respect to the baseline nn-Unet, and is particularly impressive with respect to out-of-distribution data. The authors also show that using deep SDFs for this task generates superior overlap estimates than converting the outputs to meshes.

**Weaknesses:**

- W1. From my understanding, the two major contributions are the way of regularizing the multi-organ reconstruction approach loss functions constructed through surface aware Monte Carlo sampling, and using deep SDF as a representation for this task. Numerous loss functions for regularizing with respect to surface contact have been explored over the years [1,2,3]. Some others were designed for 2D but obey the same principles of (lack of) intersection and contact as explored here.
My main concern is that the method evaluation is limited to nn-Unet and fitting the SDFs to each organ individually. There are other losses that have been used to regularize topological consistency of medical organs; the authors should compare to these, as is I don’t think the experimental aspects of this paper do justice to the previous literature on this topic.
While the authors mention the closely related method by Gupta et al [1], they discard it in the introduction as it only handles local constraints, and cannot be used to enforce global organ contact priors.  However, specifically with respect to my later point (see W2), such an approach may bias segmentations less.
- W2. Medical relevance is not explored despite being the primary motivation for this work. Enforcing a certain pre-specified level of contact between organ pairs is certainly useful for healthy patients, but in pathological cases one might specifically seek to find violations or deviations from such an overlap. The authors should at the very least mention how these losses might bias predictions towards reconstructions that mimic healthy organs. The paper would be much stronger and application relevant if this were explored.
- W3. In the introduction the authors state (L98-99) that the latent vector of the 3D SDF is used to refine the segmentation outputs of the nnUnet. However, despite showing experiments how this approach is superior, they never actually detail how this is achieved.

[1] Gupta, S., Hu, X., Kaan, J., Jin, M., Mpoy, M., Chung, K., ... & Chen, C. (2022, October). Learning topological interactions for multi-class medical image segmentation. In *ECCV.*
[2] Ganaye, P. A., Sdika, M., Triggs, B., & Benoit-Cattin, H. (2019). Removing segmentation inconsistencies with semi-supervised non-adjacency constraint. *Medical image analysis*, *58*, 101551.
[3] Reddy, Charan, Karthik Gopinath, and Herve Lombaert. "Brain tumor segmentation using topological loss in convolutional networks." (2019).

**Questions:**

- Please expand and compare the used losses with previous losses used for topologically aware segmentation in the medical imaging (and potentially other) literature. Detail why and how the specific losses proposed here are unique, and particularly effective for the task at hand. These claims should be backed up experimentally.
- The authors should comment on the potential bias such a prior induces upon outputs in pathological cases. Ideally, this would also be backed up experimentally. Could this prior be determined on a patient-specific basis, or based on other factors besides a specific pre-defined overlap?
- Please clarify the refinement of the nn-Unet segmentations using the latent SDF representation, as this part is not detailed clearly in the paper.

---

### Note · Authors · 2024-11-27

**Comment:**

We thank the reviewers for their time and feedback.

**Withdrawal Confirmation:**

I have read and agree with the venue's withdrawal policy on behalf of myself and my co-authors.